# Efficacy of *Salmonella* Bacteriophage S1 Delivered and Released by Alginate Beads in a Chicken Model of Infection

**DOI:** 10.3390/v13101932

**Published:** 2021-09-25

**Authors:** Janeth Gomez-Garcia, Alejandra Chavez-Carbajal, Nallelyt Segundo-Arizmendi, Miriam G. Baron-Pichardo, Susana E. Mendoza-Elvira, Efren Hernandez-Baltazar, Alexander P. Hynes, Oscar Torres-Angeles

**Affiliations:** 1Laboratory of Microbiology and Parasitology, School of Pharmacy, Autonomous University of the State of Morelos, 1001 University Avenue, Chamilpa, Cuernavaca 62209, Mexico; janeth.gomezgar@uaem.edu.mx (J.G.-G.); san_ff@uaem.mx (N.S.-A.); dpmg_ff@uaem.mx (M.G.B.-P.); efrenhb@uaem.mx (E.H.-B.); 2Departament of Medicine, McMaster University, 1280 Main Street West, Hamilton, ON L8S 4K1, Canada; carbajaa@mcmaster.ca; 3Laboratory of Virology Postgraduate Field 1, Cuautitlán School of Higher Studies, National Autonomous University of Mexico, 1st May Avenue, Sta María Guadalupe las Torres, Cuautitlán Izcalli 54740, Mexico; seme_6@yahoo.com.mx

**Keywords:** phage therapy, *Salmonella* enteritidis, poultry sector, alginate

## Abstract

Modern bacteriophage encapsulation methods based on polymers such as alginate have been developed recently for their use in phage therapy for veterinary purposes. In birds, it has been proven that using this delivery system allows the release of the bacteriophage in the small intestine, the site of infection by *Salmonella* spp. This work designed an approach for phage therapy using encapsulation by ionotropic gelation of the lytic bacteriophage S1 for *Salmonella* *enterica* in 2% *w*/*v* alginate beads using 2% *w*/*v* calcium chloride as crosslinking agent. This formulation resulted in beads with an average size of 3.73 ± 0.04 mm and an encapsulation efficiency of 70%. In vitro, the beads protected the bacteriophages from pH 3 and released them at higher pH. To confirm that this would protect the bacteriophages from gastrointestinal pH changes, we tested the phage infectivity in vivo assay. Using a model chicken (*Gallus gallus domesticus*) infected with *Salmonella* Enteritidis, we confirmed that after 3 h of the beads delivery, infective phages were present in the chicken’s duodenal and caecal sections. This study demonstrates that our phage formulation is an effective system for release and delivery of bacteriophage S1 against *Salmonella* Enteritidis with potential use in the poultry sector.

## 1. Introduction

Bacteria of the genus *Salmonella* can infect humans and animals. Two of the main pathogens of these genera infecting animals, and specifically poultry, are *Salmonella enterica* serovar Pollurum (*S.* Pullorum) and *Salmonella enterica* serovar Gallinarum (*S.* Gallinarum) [1]. Their economic impact in terms of loss in the poultry sector is due to their killing effect after infection. Both strains can cause death in up to 90% of birds affected [1,2]. Another example is *Salmonella enterica* serovar Enteritidis (*S.* Enteritidis), which can infect humans through consumption of contaminated meat and eggs. At the same time, poultry infected with *S.* Enteritidis infect their healthy neighbors contributing to further economic loss. *S.* Enteritidis is one of the main foodborne pathogens responsible for spreading *Salmonella* in humans, representing a public health issue [3,4]. The indiscriminate use of antibiotics as growth promoters for weight gain in poultry contributed to the spread of antibiotic-resistant microorganisms in humans and has prompted the prohibition of antibiotics in the poultry sector, so bacteriophages or phages (viruses that infect and lyse bacteria) have emerged as a strategy bio-control of *Salmonella*. This can include applications intended to limit the spread of the bacterium or directly for the treatment of salmonellosis in poultry, an approach known as phage therapy [5,6,7,8,9,10,11,12].

Currently, a wide variety of studies on phage therapy in the poultry sector demonstrated that its use is effective in reducing bacterial loads of *Salmonella* that colonize the digestive tract [13,14,15]. In one example, Fiorentin et al. [16] demonstrated the effectiveness of a cocktail of lytic phages to reduce *Salmonella* Enteritidis in cecal tonsils of day-old chickens. Five days after treatment, *Salmonella* Enteritidis loads decreased by 3.5 orders of magnitude. Andreatti-Filho et al. [17] demonstrated that the oral administration of a cocktail of selected phages helped to reduce the colonization of *Salmonella* Enteritidis in cecal tonsils during the first 24 h. However, the treatment failed to achieve a significant reduction after 48 h. Despite these reported successes of the solutions and cocktails, they have limited stability due to loss of the phage titer during processing and storage, which in a few months could represent losses as high as 99% of the initial titer.

In fact, one of the main challenges of phage delivery forms has been their stability, which is why different authors have been inclined to look for solid pharmaceutical options such as nanosomes, liposomes or microspheres [18]. Currently, alginate (ALG) is the main biopolymer used to encapsulate phages due to its biocompatibility, low price and low toxicity [19]. One of the first reports about the encapsulation of bacteriophages in ALG using the ionic gelation technique with calcium as a cross-linking agent was published in 2008 by Yongsheng et al. [20], who encapsulated the *Salmonella* spp. phage Felix O1 in a chitosan-alginate- calcium chloride system, demonstrating that the ALG allowed Felix O1 to be infective in the gastrointestinal tract (GIT), promoting the phage delivery in their intestines.

We previously isolated and partially characterized a phage, “S1”, capable of infecting *Salmonella enterica* in vitro [21]. Briefly, phage S1 has a plaque morphology consistent with a lytic life cycle, a burst size of 28–40, and is resistant to temperatures up to 50 °C. For these reasons, phage S1 was considered a candidate for phage therapy in poultry with the goal of preventing the transmission of *Salmonella enterica* to humans.

In order to use S1 in phage therapy, a thorough genetic characterization of the phage is required to ensure it is virulent (strictly lytic) and carries no genes that would be a barrier to its use; furthermore, as the gastrointestinal pH of chickens varies from 1 to 4 in the gizzards, while ranging from 5 to 7 in the duodenum, the phage would experience a wide-range of pHs which could impair its efficacy [22,23]. For this reason, the phage might require a delivery vehicle in order to be effective in therapeutic applications [19]. Accordingly, we set out to complete the genetic characterization of the phage, develop a suitable formulation to encapsulate S1, and assess the delivery and release of phage S1 in vitro and in vivo models.

## 2. Materials and Method

### 2.1. Bacterial Strains

A clinical poultry isolate of *Salmonella enterica* serovar Enteritidis phage type 13A (SE PT13A), was obtained from the USDA National Veterinary Services Laboratory (Ames, IA, USA) and donated by Dr. Billy Hargis and Dr. Guillermo Tellez of the Department of Poultry Science, University of Arkansas. SE PT13A was grown on tryptic soy agar (TSA) (BD Bioxon, Mexico City, Mexico) or tryptic soy broth (TSB) (BD Bioxon, Mexico City, Mexico) according to the assay needs. Strains *Salmonella enterica* serovar Pullorum (MDR-MC862-A) and *Salmonella enterica* serovar Gallinarum (MDR-MC862-B), were isolated from clinical trials in poultry and donated by the Faculty of Veterinary, National Autonomous University of Mexico. The strain *Salmonella enterica* serovar Cholerasuis (*S.* Cholerasuis), *Salmonella enterica* serovar Typhi (*S.* Typhi), *Salmonella enterica* serovar Typhimurium (*S.* Typhimurium) and *Citrobacter freundii* (*C. freundii*), *Enterobacter cloacae* (*E. cloacae*) and *Escherichia coli* (*E. coli*) were isolated from clinical trials in humans (Appendix A).

### 2.2. Phage S1 Isolation

Phage S1 was isolated from wastewater from the city of Cuernavaca Morelos, Mexico [21]. Propagation of phage SI was carried out according to what was reported by Segundo-Arizmendi et al. [21], when the host bacterial culture SE PT13A reached mid-exponential phase in TSB a 37 °C and 120 rpm, (0.2A with OD630 nm) the culture was infected with phage S1 at multiplicity of infection (MOI) of 1 and incubated for a further 5 h. The culture was then centrifuged at 1300× *g* for 15 min (Centrificient V1, CRM GLOBE). The supernatant was filtered by passage through 0.45-µm cellulose acetate membrane filter (MF-Millipore, Merck KGaA, Burlington, MA, USA) and stored at 4 °C.

### 2.3. Enumeration of Phages

The enumeration of phage S1 was performed using the double-layer agar method described by Adams [24], with some modifications. An aliquot of 100 µL of mid-exponential SE PT13A was added to 100 µL of ten-fold serial dilutions of phage S1 with sterile saline solution 0.85% *w*/*v* (NaCl ACS Fermont, Monterrey, Nuevo León, Mexico) to a tube containing 3 mL of bacteriological agar at 0.7% *w*/*v* and was poured in triplicate into TSA plates that were incubated at 37 °C for 24 h.

### 2.4. Sequencing and Annotation of Phage S1

After obtaining a high titer (>1 × 10^11^ PFU/mL) of phage S1, a phage DNA extraction was performed with Invitrogen PureLink Viral RNA/DNA Mini Kit (Thermo Fisher Scientific HR Services Mexico, S. de R.L. de C.V. Mexico, Mexico.) following the kit protocol. Library preparation of the resultant DNA was performed by tagmentation using Illumina Nextera protocol. The final library was prepared for sequencing with the reagent MiSeq kit v2 (Illumina Inc, San Diego, CA, USA 92122, EE.UU.) and sequenced using Illumina MiSeq System. The fragments obtained were paired-end reads with lengths of 150 bp. Host reads and adapters were trimmed using Trimmomatic v0.36 [25] resulting in more than 8 million paired-end sequence reads. De novo assembly was carried out with ABySS v2.0.1 [26] using default parameters. To determine similarities of the assembled genome among all genomic sequences we used BLASTn [27]. Genome annotation was performed with PROKKA v1.14.6 [28]. Phage annotation was performed by HMMER v 3.3.2 [29] to search phage protein similarities against UniProt database. Alternately, an alignment of the closest phage S1 relatives *Salmonella* phage ZCSE2 (NC_048179) and S144 (MT663719) was used to identify protein hits and their *e*-values. For each protein comparisons the *e*-value from the full sequence was taken into consideration. Additional protein searching was made with the open reading frames (ORF’s) using HHpred toolkit [30]. To classify the phylogeny of S1, we used ViPTree to generate the viral proteomic tree [31]. Three sequences from the *Loughboroughvirus* genus (*Salmonella* phage ZCSE2; NC_048179, phage S144; MT663719 and phage SE4; NC_048764) were uploaded into the web server to generate the phylogenetic tree. The completeness of the final assembly was confirmed by mapping the sequences from the last and first portions of S1 genome and compared with the linear assembly. The obtained tandem repeat was visualized by Tablet v.1.17.08.17 [32] for quality control. The manual curation and final annotation were done by Artemis v18.1.0 [33]. Blastp was used to identify the non-redundant protein sequences. DNAplotter [34] was used to obtain the circular organization from the final genome annotation. An alignment of protein coding genes among phages with >87% of nucleotide identity was built up using a Python script [35]. Additionally, predictive analysis to identify the phage lifecycle was made using PhageAI [36]. The sequence was deposited in the NCBI repository under the accession number: MZ127825.

### 2.5. Efficiency of Plating of Phage S1

The efficiency of plating (EOP) of phage S1 was determined in five serovars of *Salmonella enterica* (*S.* Gallinarum, *S.* Pullorum, *S.* Typhi, *S.* Typhimurium and *S.* Cholerasuis). Additionally, the EOP was tested in other gram-negative species; *C. freundii*, *E. cloacae* and *E. coli* (Appendix A). The EOP was done according to the technique described by Montso et al. [37] with some modifications. The strains were adjusted at a set titre of 1.6 × 10^8^ CFU/mL, while ten-fold serial dilutions of phage S1 were made with sterile saline solution 0.85% *w/v* until obtaining 1 × 10^2^ PFU/mL. Then, 100 µL of phage S1 was mixed with 100 µL of each strain during 10 min at room temperature. The mixture was added into 3 mL of bacteriological agar at 0.7% *w/v* and poured on top of TSA plates and incubated at 37 °C for 24 h. The relative EOP of phage S1 was determined by equation 1, from the average of five trials.
(1)$$Reative\;EOP=\frac{average\;number\;of\;plaques\;on\;targeted\;host\;bacterium\;\left(PFU^{\prime }s\right)}{average\;number\;of\;plaques\;on\;reference\;host\;bacterium\;\left(PFU^{\prime }s\right)}$$

The EOP value obtained was classified as high (EOP ≥ 0.5), medium (0.5 ≤ EOP > 0.01), and low (EOP ≤ 0.1) [34].

### 2.6. Encapsulation of Phage S1 in ALG Beads

Bacteriophage S1 was encapsulated using the ionic gelation technique by extrusion [12,19,38]. A solution of ALG (Drogueria Cosmopolita, México city, Mexico) was prepared at a concentration of 2% *w*/*v* and 1.19 × 10^11^ PFU of phage S1 was added, the mixture was extruded using a 5 mm internal diameter needle to a sterile calcium chloride solution 2% *w*/*v* (Fermont, Monterrey, Nuevo León, Mexico). To remove excess calcium chloride solution from the beads, three washes were made with 150 mL of sterile deionized water for 3 min and kept at room temperature for two hours to remove excess water. Finally, 100 beads were measured with a digital calibrator UltraTech^®^ (General Tools, New York, NY, USA) and stored at 4 °C.

### 2.7. Determination of the Encapsulation Efficiency

The encapsulation efficiency (%EE) was determined according to what was reported by Colom et al. [38] and Boggione et al. [39] with modifications. The encapsulated phage S1 was recovered from beads inoculated in denominated broken microsphere solution containing 50 mM sodium citrate (Fermont, Monterrey, Nuevo León, Mexico) and 0.2 M sodium bicarbonate (Fermont, Monterrey, Nuevo León, Mexico) adjusted to pH 7.5 with HCl 0.2 M (Fermont, Monterrey, Nuevo León, Mexico) and shaken at 120 rpm until macroscopic and visible disintegration for the liberation of the phage. The amount of phage S1 that not encapsulated during the alginate ion gelling process was determinate of the calcium chloride solution and the bead wash water. Finally, ten-fold serial dilutions of the broken microsphere solution, calcium chloride solution and the water for washing the beads were made on sterile saline solution 0.85% *w*/*v*. The determination of the total PFU was made using the double layer agar method by Adams [24]. The %EE of phage S1 was calculated as the percentage of phage encapsulated within the beads compared to the total phage titer and was determined using equation 2 reported by González-Menéndez et al. [37]
(2)$$\%EE\frac{encapsulated\;phage\;\left(PFU\right)}{total\;phage\;\left(PFU\right)}\;\times \;100$$
where the encapsulated phage is the amount of PFU of phage S1 recovered from beads and total phage is the sum of PFU recovered from beads and PFU of non-encapsulated.

### 2.8. Comparison of SE PT13A Growth under the Exposure of ALG and ALG + Phage

SE PT13A cultures were grown to the middle of the exponential growth phase, then inoculated with 1 g of alginate beads with and without phage S1. The cultures were maintained at 37 °C with constant agitation at 120 rpm. Samples were taken at 0, 12, 24, 36 and 48 h after inoculation and ten-fold dilutions were performed with sterile saline solution 0.85% *w*/*v*. An aliquot of 500 µL of each dilution per triplicate was seeded in TSA plates by spatulation technique and incubated at 37 °C for 24 h. The bacterial titer (CFU/mL) was calculated according to equation 3 as follows:(3)$$\frac{CFU}{mL}=\left[number\;of\;colonies\right]\left[dilution\;factor\right]\left[\frac{1}{aliquot}\right]$$

### 2.9. In Vitro Protection and Telease of Free Phages and Encapsulated Phages in ALG Beads

To determine whether the alginate beads protect bacteriophage S1, we designed the following experiment. Free phages at a fixed concentration of 9.56 × 10^8^ PFU were mixed on 10 mL of sterile saline solution 0.85% *w/v*, the final pH was adjusted to 3, 5, 7 and 8.5 with HCl 0.2 M and NaOH 0.2 M (Fermont, Monterrey, Nuevo León, México), accordingly. The mixture was incubated at 40 °C and in agitation for 100 rpm, aiming to mimic the average temperature of the chicken GIT, and at either pH 3 for 45 min, pH 5 for 15 min, pH 7 for 30 min, and pH 8.5 for 20 min, aiming to mimic the pH and transit time of the proventriculus + gizzard, duodenum, jejunum + ileum or cecum, respectively [22,23,40]. The free phages maintained at pH 3 and 5 were neutralized with a 0.2 M NaOH solution for 5 min and placed in 5 mL of sterile saline solution at pH 7 in order not to affect bacterial viability at the time of titration of the phage, as *Salmonella* grows in an optimum pH of 6.5 to 7.5 and its growth is inhibited at pH < 3.8 [41].

On the other hand, 1 g of ALG beads containing the phage were incubated under the same conditions as the free phages described above. After incubation, the beads on pH 3 and 5 were adjusted at pH 7 with 0.2 M NaOH for 5 min and rinsed with sterile distilled water three times and placed in 20 mL of the broken microsphere solution at pH 7.5 and stirred at 120 rpm until macroscopic and visible disintegration for the release of the phage [20] as the ALG beads disintegrate at pH >7. Then the beads on pH 7 and 8.5 were rinsed with sterile distilled water and placed in 20 mL of the broken microsphere solution at pH 7.5 and stirred at 120 rpm until macroscopic and visible disintegration for the release of the phage. Each sample was titered as described previously by the Adams technique [24]. The PFU/g were calculated using the PFU/mL multiplied by 20 mL of the broken microsphere solution used for the beads disintegration. The percentage of phage recovery was calculated using the titers from the initial phage concentration at time zero.

### 2.10. In Vivo Assays

#### 2.10.1. Experimental Animals

A sample size of 38 one-month-old chickens was calculated taking into consideration 5% of chicken loss during the transportation. We obtained all animals from a commercial supplier (Comercializadora Granja Cuevas, Tolteca, Mexico). The Institutional Project Review Board “Ad hoc Committee for admission to Doctorate in Pharmacy” assigned by Faculty of Pharmacy, UAEM approved the ID# 2019/003 research protocol in 17 June 2019. Chickens were transferred and handled in accordance with the Official Mexican Standard NOM-051-ZOO-1995, Humanitarian treatment in the mobilization of animals [42]. After one week of observation after they arrived, the chickens were used to perform the in vivo experiments. The animals were kept in the School of Pharmacy, UAEM, under the specifications of the facilities established in Guide for the Care and Use of Agricultural Animals in Research and Teaching, 3rd edition. [43] and the Manual of Good Livestock Practices in Broiler Production Units, 2nd edition [44]. To guarantee that the chickens were free of *Salmonella*, upon arrival, samples were taken from the cloaca with sterile swabs that were sown in TSB and incubated at 37 °C for 24 h. The animals were kept in the conditioning phase for seven days by supplying them with commercial growth food FlagasaTM (Azcapotzalco, Mexico) and water ad libitum. Thirty chickens were distributed in five groups for elimination test and eight were used for the identification of the degradation of the ALG beads and phage S1 release test. Finally, the animals were euthanized by cervical dislocation in accordance with the 4th edition of the Guide for the Care and Use of Agricultural Animals in Research and Teaching [43] Chapter 2: Agricultural Animal Health Care and by the American Veterinary Medical Association and the provisions of the Official Mexican Standard NOM-033-SAG/ZOO-2014; Methods for killing domestic and wild animals [45].

#### 2.10.2. Degradation of the ALG Beads and Phage S1 Release

To observe the degradation process of the ALG beads during their passage through the gastrointestinal tract, a total of eight chickens were administered 1 g of alginate beads containing 4.9 × 10^9^ PFU phage S1. Two randomly chosen chickens were sacrificed at 1, 3, 5 and 18 h after administration and macroscopic search for ALG beads was performed in crop, gizzard, cecum and duodenum cavities of the GIT. In addition, the presence of phage S1 was determined as follows: the contents of each of the mentioned cavities was homogenized in 20 mL of sterile saline solution 0.5% *w/v* and centrifuged at 1300× *g* for 15 min, the supernatant was filtered using a 0.45-µm filter. An aliquot of 50 µL from that filtrate was inoculated on TSA plates which contained a lawn of SE PT13A (spot test technique). Plates were incubated at 37 °C for 24 h. The presence of lytic zones was considered as a positive for phage activity in each of the analyzed chicken organs.

#### 2.10.3. Elimination of SE PT13A in Chickens

Knowing that phages are biological agents that replicate in the host cell, we designed an assay that would corroborate their presence in organs that *Salmonella* spp. infects. The chickens’ infection with SE PT13A and administration of the corresponding treatments were performed as indicated in Table 1. Post infection day three, five, and seven samples were taken with swabs from the cloacae in each group. Then, they were seeded in Petri dishes containing TSA and incubated at 37 °C for 24 h. One-day post-infection (Day 8) with SE PT13A, each chicken received a single dose of the treatment indicated for their group (one gram of beads with or without phage, or antibiotic). The reduction of SE PT13A was determined by qualitative analysis of the bacterial growth of the cloacal samples, obtained from each group 24 and 48 h after treatment administration, on TSA at 37 °C for 24 h. Additionally, 10 g of stool sample obtained from the chicken beds in groups 1 and 4 were considered for phage screening after 24 h of the treatment with ALG beads. Stool samples were hydrated in 10 mL of sterile saline and centrifuged at 1300× *g* for 15 min. The supernatant was analyzed by spot test as described previously, to determine if the encapsulated phage S1 had passed through the GIT, been released, and remained infectious.

## 3. Results and Discussion

### 3.1. Sequencing of Phage S1

We obtained a complete and novel assembled phage genome of 53, 394 base pairs in length, with an average coverage of 7000× and genomic guanine and cytosine (G + C) content of 45.8%. The genome annotation of phage S1 resulted in 75 ORF’s, while the prediction of the protein functions assigned only 29 ORF’s (Table 2 from the NCBI database, HMMER or HHpred at the time of the search (26 July 2021).

S1 has a core of the main proteins proteins expected in its genome; genes encoding major capsid proteins, phage tail fibers, lysis and replication (DNA polymerase I and ATP-dependent RNA helicase) and phage terminase proteins. We did not detect tRNAs, transposases, integrases or phage repressors. A schematic gene organization is shown in Figure 1.

To predict its lifecycle, we used a predictive analysis tool (PhageAI), which anticipated phage S1 as 100% virulent. We confirmed the completeness of the phage S1 after mapping their reads from the end and beginning of its linear sequence.

The coverage of these tandem repeats resulted in an average 50×. However, we couldn’t determine the phage termini due the tagmentation protocol used in the library preparation.

Nucleotide analyses on BLASTn shown that the genomic content from phage S1 shared the highest percentage of identity with three *Salmonella* phages; S144 (97.64%), ZCSE2 (97.58%) and SE4 (87.05%), the accession numbers are NC_048179, MT663719 and NC_048764 respectively. These three phages belong to the family *Myoviridae* and genus *Loughboroughvirus*. We carried out a phylogenetic analysis using a full viral protein alignment to classify S1 through the database from ViPTree. Phage sequences from ZCSE2 and S144 are considered relatively new. The release of these sequences to the NCBI database happened during 2020. For that reason, we added the new genomes into the webserver to identify the phylogeny of S1. The proteomic tree showed that S1 had the closest relationship with phage ZCSE2, suggesting that S1 comes from the same linage, the genus *Loughboroughvirus*. Also, we denoted a second cluster showed a close relationship between ZCSE2 and S144 and a third cluster showing the relation of SE4 with the rest of phages from the same genus (Figure 2).

Phage S144 is the closest phage related to S1 with 97.64% nucleotide identity over the length of its genome. From the 75 proteins in S1, only 70 were found in S1 with 85% amino acid similarity cutoff (Figure 3).

Three proteins with the open reading frame in the complementary strand were not more than 85% similar among phages from the same genus. From these proteins, just two had a predictive regulatory function.

A remarkable characteristic of phage S144 is the fact that it can infect 25 *Salmonella* serovars but also can infect at least 4 *Cronobacter sakazakii* strains and *Enterobacter cloacae*, with complete different plaque morphology in different hosts. This broad host infectivity positioned the phage S144 as a polyvalent phage [46]. In our study, we did not detect polyvalence. Nonetheless, it is interesting that within the less than 2.3% of dissimilarity at the nucleotide level exists impactful differences across this new genus.

Phage ZCSE2 it is a mere 571 bp shorter than ZCSE2 and 97.58% identical over the entire length of its genome arising two gaps at the beginning and end of their sequences. S1 encodes 75 proteins, all but 73 of which can be found in S1 as well (85% amino acid similarity cutoff). We noticed that the ORFs 50 and 56 in phage S1 are unique proteins that are not present in any other phage from the same genus (at more than 85% amino acid similarity cutoff). These are small proteins with 50 and 58 amino acids, respectively (Figure 3), with their open reading frame in the complementary strand with unknown function even after the predictive searching with HHpred. ZCSE2 has a broad host range against 24 *Salmonella* strains representing 16 serotypes and is stable in a high titer concentration on an extensive pH spectrum (4.5–9 pH) up to 24 h [47]. These features highlight the importance of the discovery of other phages as S1 with high similarity to ZCSE2, considering these features that allow fighting against pathogens that are potentially harmful to humans and animals.

A phage less closely related to S1 but from still the same genus is SE4, which is 100 bp longer than S1 and 87.05% identical over the entire length of its genome. From the 75 proteins in S1, only 57 were found in S1 and in the others with 85% amino acid similarity cutoff (Figure 3). The majority of these 57 proteins are hypothetical proteins. Only four of them had annotated functions in their proteins on the ORFs 36, 49, 50 and 63. The first three proteins had functions implicated in tail fibers, major tail and tail assembly, respectively, and are in the same location of the proteins involved in morphogenesis than the other phages. But in the fourth protein (ORF 63), we identified an annotated protein that codifies to an enzyme involved in the pathway of galactose metabolism, the UDP-Glucose 4-epimerase. These four ORFs together may indicate a remarkable and unique difference among SE4 and the rest phages from the same genus. Not less important than ZCSE2 and S144, the host range of SE4 is extensive as well, leading to infect 36 different strains of *Salmonella enterica*, indicating that SE4 also has a broad spectrum of bacterial killing [48] and high potential to use against a wide range of *Salmonella* spp.

The conserved proteins across their phylogenetic neighbors is shown in Figure 4, in which the alignment of the proteins set at 95% cutoff of the amino acid similarity. The ORF 38 in phage S1 correspond at a putative exonuclease and ORFs 50, 56, 66 and 70 have unknown functions but are different across the rest of the phages.

### 3.2. Host Range by EOP

The EOP of phage S1 in *S.* Pullorum MDR-MC862-A was 0.61 while with *S.* Gallinarum MDR-MC862-B was 0.58 (Table 3). The type of plaques in both strains were clear; however, we could not see plaques as a proof of infectivity on *S*. Typhi, *S.* Typhimurium, *S.* Choleraesuis, nor on gram-negative isolates from humans. Although the subspecies of *Salmonella enterica* share a common bacterial ancestor [49], the presence of lytic plaques on *S.* Pullorum and *S.* Gallinarum was associated with a close phylogenetic relationship because these serotypes share a direct bacterial ancestor with Enteritidis serotype [50,51].

### 3.3. Particle Size and Encapsulation Efficiency

The phage S1 encapsulated by ionic gelation in ALG beads using calcium chloride as a crosslinking agent resulted in an average size of 3.73 mm with a range of 3.69 to 3.77 mm. The size of the beads formed by the ionic gelation technique by extrusion depends mainly on the diameter of the nozzle used, which is the determining factor for obtaining the desired size [52]. Considering that the pressure exerted with the nozzle increased when the diameter of the nozzle decreased [53], and at higher pressure tailed phages could release their genetic material, resulting in loss of infectivity [12], we used a nozzle with an internal diameter of 5 mm. In this work, the size of ALG beads used to encapsulate the bacteriophage S1 are higher than in other studies [20,39,54]. The bead size reported by Abdelsattar et al. [55], was 2.38 ± 0.14, 2.8 ± 0.11 and 2.33 ± 0.12 mm for three different formulations of the bacteriophage ZCEC5. However, our formulation’s bead size can be advantageous because increasing the particle size of the chicken food improves gizzard functionality and development [56,57,58]. On the other hand, the assessment of the percentage of encapsulation efficiency showed an average of 88.09% (1.11 × 10^11^ PFU). This encapsulation percentage was slightly less than reported in other studies where the beads disintegrated under the same conditions, such as that by Moghtader et al. [59], who reported an efficiency of 90% in the encapsulation of bacteriophage T4 in 2% *w/v* alginate beads coated with chitosan and polyethyleneimine, as well as the study by Yongsheng et al. [20], who obtained an efficiency of 93.3% when encapsulating the phage Felix O1 in alginate microbeads coated with chitosan. However, it is important to highlight that our formulation does not contain other additional polymers like chitosan or polyethylenimine. Also, several products based on phages are traded in diverse preparations such as aerosol lyophilizes or nasal sprays, pills, creams, ointments and liquid preparations with titers from 10^5^ up to 10^11^ [60,61]. For these reasons, we consider that the PFU of bacteriophage S1 contained in the ALG beads is suitable for their implementation as phage therapy.

### 3.4. Comparison of SE PT13A Growth under the Exposure of ALG and ALG + Phage

We first set out to ensure the alginate beads would not interfere with our ability to detect phage activity in vitro. We compared the bacterial growth exposing the ALG beads and ALG beads + phage. When SE PT13A was grown in the presence of ALG beads, the viable bacterial counts increased X-fold over the first 24 h, with a drop of less than 2-fold over the following 24 h (Figure 5), consistent with standard exponential growth and eventual ‘death phase’. This is consistent with the fact that no previous study has reported any antimicrobial properties of alginate. In contrast, when SE PT13A was grown in presence of ALG + phages, bacterial growth was drastically inhibited (Figure 5), with this effect persisting for the full 48 h, with no increase in host growth that would be expected from escape mutants. The presence of ALG blocks neither our ability to detect phage activity, nor the activity itself.

### 3.5. In Vitro Protection and Release of Free Phages and Encapsulated Phages in ALG Beads

The alginate bonds are strengthened when they are in the presence of acidic pH values. For this reason, the pH is relevant to preserve ALG beads made by ionic gelation. As more basic pH, the beads depolymerize, resulting in the delivery of phages [62,63]. In contrast, at more acidic pH, polymers favour the formation of higher amounts of hydrogen bonds in the polymeric network, which prevents the delivery of the active ingredient [64]. Knowing this, we aimed to test the phage protection and release on ALG beads mimicking the pH and the gastrointestinal transit time of each GIT section of the chickens. For that, we exposed the phage S1 to pH 3 for 45 min, pH 5 for 15 min, pH 7 for 30 min and pH 8.5 for 20 min, followed by neutralization with NaOH (0.2M) to identify the percentage of phage recovery as described in the methods section. Figure 6 shows that at a pH of 3; the number of phages recovered after neutralization was 38.27%. While the phage encapsulated in ALG after neutralization and breaking the beads, the recovered phage was 85.23%, suggesting that the ALG beads prevent the release of the encapsulated virus in the polymeric matrix, favouring its protection at this pH. At pH of 5; the release conditions did not change regardless of whether the virus was free or encapsulated. In contrast, at pH 7 and 8.5 where the beads were expected to begin to release phages (Figure 6, Right), the encapsulated phages showed a recovery percentage up to 60%, while we obtained 90.08% of recovery with the free phages. Together, this suggests that ~40% of the phages had been released from the beads in this time—indicative of the ALG beads successfully release—albeit with some delay—phage at these pH values (see Appendix A). This agrees with Hjorth and Karlsen [62], who indicate that the beads undergo a process of hydration, dissolution and erosion for the release of the active ingredients. The encapsulation of bacteriophages in ALG demonstrated to prevent bacteriophages from being released at acidic pH values where its host is not likely living and enable the phage release at an appropriate pH of the GIT where *Salmonella* lives, from duodenum to the cloacal zone, where the pH ranges from 6 to 8 on average [65]. This is relevant because it shows that the ALG beads are capable of preserving the biological activity of the virus at low pH values, where it is likely where the bacteria wouldn’t be colonizing, but is released where the pH is optimal for its host. We conclude that the formulation allows the release of the phage in a controlled manner at pH values where SEPT13A could be colonizing the gastrointestinal tract, thus prolonging their release.

### 3.6. In Vivo Assays

#### 3.6.1. Degradation of the ALG Beads and Phage S1 Release

The passage of the beads in the GIT of the chicken was visualized macroscopically when dissecting two animals at random at 1, 3, 5, and 17 h after administration. It was only possible to observe the alginate beads in the crop of chickens sacrificed 1 h after administration. These results are in agreement with the literature, indicating that there exists a decrease in particle size of chicken feed caused by the process of crushing and maceration by the gizzard [66,67]. Besides, Amerah et al. [56] determined that the food should have a particle size < 0.1 mm to reach the duodenum. One of the main concerns about triturating the ALG beads is the release and exposure of the phage S1 into the gizzards’ pH (2.5–3.5) due to their sensitivity at pH < 3. For this reason, to ensure that phage S1 was infective in the duodenum and cecum after the beads passing through the gizzard, we performed an in house beads test which is summarized in Table 4. We obtained plaques from the chicken crops at 1, 3 and 5 h. Neither of the samples showed lytic plaques in the gizzards but, 50% of the sacrificed chickens at 3 and 5 h showed lytic plaques in the duodenum. Finally, 100% of the chickens at 3 h and 50% at 5 h showed lytic plaques in the cecum samples. These results suggest that even after the beads’ trituration by the chicken gizzards’, our formulation at 2% *w/v* of ALG beads protects the phage when it is exposed at acidic pH, keeping it infective in the duodenum and cecum, cavities that are infected by the *Salmonella* genus.

#### 3.6.2. Elimination of SE PT13A in Chickens

By ensuring the absence of the genus *Salmonella* in the experimental animals and after the conditioning phase through cloacal swabs, the corresponding infection was performed in each group. *Salmonella* colonizes the GIT of birds, mainly the cecum and cloaca. The incubation period of *Salmonella* infection in poultry varies from four to seven days, so the cloacal samplings obtained at three, five, and seven days after infection with SE PT13A showed that at days three and five, in all groups, the presence of lactose-positive, nonproducing hydrogen sulfide colonies, not consistent with the biochemical characteristics of SE in the TSA, were observed [68]. Finally, on day seven, groups 2 to 5 showed growth of circular, black colonies due to the production of hydrogen sulfide, consistent with the characteristics of the genus SE PT13A in TSA. Once the infection was confirmed, the treatment corresponding to each group was administered. The reduction in SE PT13A was determined in the cloacal samples obtained 24 and 48 h after the administration of the corresponding treatment. Figure 7 shows the elimination of SE PT13A in the five groups. Group A was the control group, which consisted of uninfected and untreated chickens. Group B was the group infected with SE PT13A without treatment. Group C were chickens infected with the bacteria and treated with antibiotics. Group D consisted of chickens infected with SE PT13A and treated with 1 g of ALG beads containing the encapsulated phages. And group E consisted of infected chickens treated with ALG beads without phages. Our results indicate that the alginate beads containing phage S1 resulted in the elimination of SE PT13A in the *Gallus gallus domesticus* model. Group C had a visibly decreased bacterial growth compared with group D. The results obtained are consistent with those reported by Andreatti Filho et al. [17] who evaluated the reduction in SE by two unencapsulated bacteriophages separately and in a cocktail, where at 48 h after treatment in cecum samples, the growth of SE was not observed in any group. Colom et al. [38] measured the in vivo reduction in *S.* Typhimurium by a cocktail of phages encapsulated in liposomes and detected a significant reduction in the cecum of 3.8, 3.9, and 1.5 logarithmic units of *S.* Typhimurium on days eight, 10 and 15 posttreatment, respectively. Likewise, Bardina et al. [69] administered a solution of phage cocktail against *S.* Typhimurium twice a day on days four and five after infection, which reduced the bacterium by one logarithmic unit on days five and six after treatment. This reduction in bacterial loads and shedding of phages could also have further benefits, as it could actively lower *Salmonella* loads in the environment, reducing the potential for feedback contamination [70,71].

## 4. Conclusions

Bacteriophage S1 is a virulent phage lacking any genes that would prevent its use in therapy, and is capable of lysing three of five strains of *Salmonella enterica* tested. Ionic gelation was an efficient encapsulation process, conferred protection of the phage from low pHs, and enabled the release of infectious particles capable of preventing *Salmonella* growth both in vitro and in vivo. Further work quantifying in vivo release and efficacy would be warranted to enable regular therapeutic or prophylactic use of encapsulated phage S1 in this context.

## Figures and Tables

**Figure 1 viruses-13-01932-f001:**
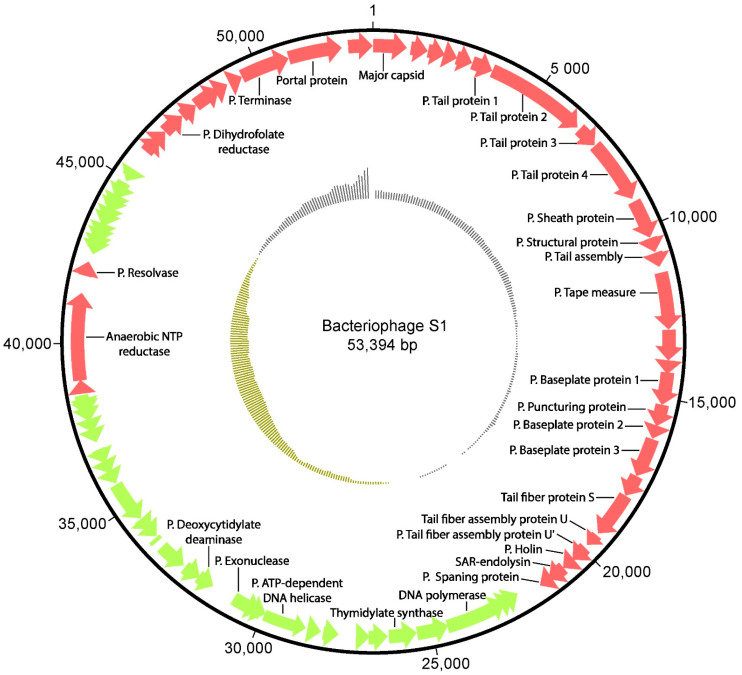
Genome representation of phage proteins with putative and known functions. The outside of the circle represents the positions of the protein-coding genes and arrow shows the direction they are transcribed. The inside of the cirle represents plots of the GC content.

**Figure 2 viruses-13-01932-f002:**
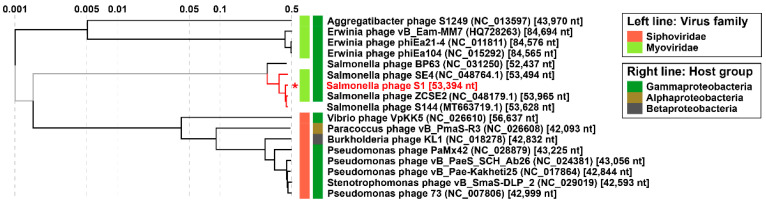
Phylogenetic tree. Phylogenetic analyses constructed by full viral protein alignment of the closest relatives phage sequences on ViPTree database and new phages from the genus *Loughboroughvirus*. Red lines indicated phages from genus *Loughboroughvirus*. And a red asterisk indicates where the phage S1 is positioned inside of the tree for easy visualization.

**Figure 3 viruses-13-01932-f003:**
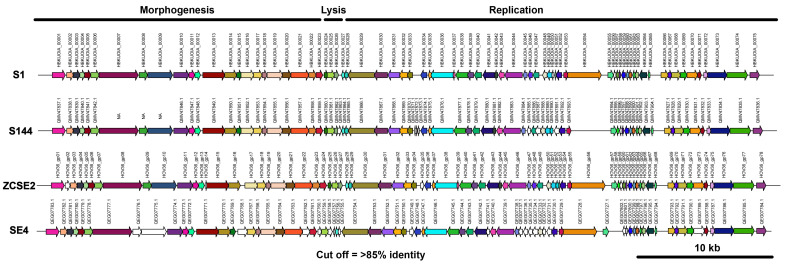
Genome alignment of viral proteins sharing >85% identity in phages from the genus *Loughboroughvirus*. From top to the bottom, the figures show phage S1, ZCSE2, S144 and SE4. ORF’s are shown with arrows and the number of each ORF is indicated on top of the arrow. Arrows with the same colors among all of them represent homologous proteins sharing >85% identity, otherwise they are shown in white. Arrows to the right indicate forward sense, arrows to the left indicate reverse sense.

**Figure 4 viruses-13-01932-f004:**
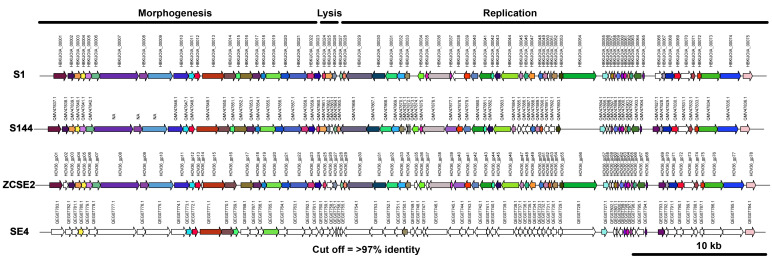
Genome alignment of viral proteins sharing >97% identity in phages from the genus *Loughboroughvirus*. From top to the bottom figures shows phage S1, S144, ZCSE2 and SE4. ORFs are shown with arrows and the number of each ORF is indicated on top of the arrow. Arrows with the same colors among all phage genomes represents homologous proteins sharing >97% identity. White arrows show <97% identity with any other protein. Arrows to the right indicate forward sense and arrows to the left indicate reverse sense.

**Figure 5 viruses-13-01932-f005:**
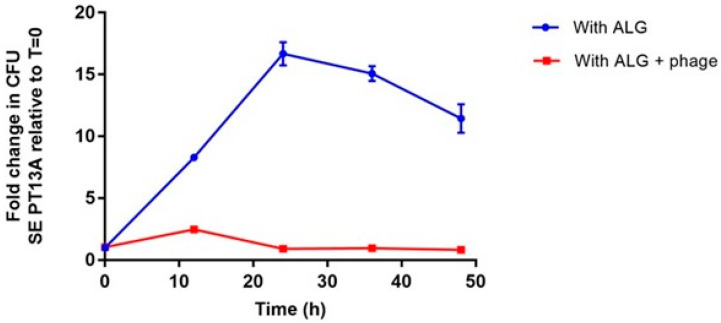
In vitro assay showing the relative influence of ALG and ALG + phage beads on the growth of *Salmonella* Enteritidis PT13A over 48 h.

**Figure 6 viruses-13-01932-f006:**
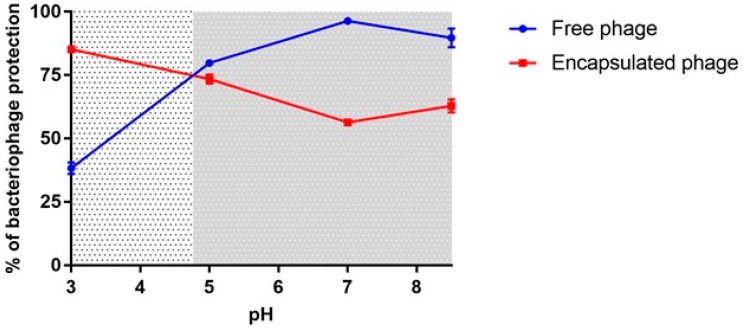
Phage recovery from exposure to pH mimicking in vivo conditions. The dotted shading (left) represents values where the beads would be expected to protect the phages and therefore high recovery indicates a high level of protection, while the grey shading (right) represents pHs where the ALG matrix would be expected to release phages, so a low recovery would indicate a high level of release. Phages were exposed to pH according to the transit time of the anatomical sections of the chicken GIT (proventriculus + gizzard at pH 3 for 45 min, duodenum at pH 5 for 15 min, jejunum + ileum at pH 7 for 30 min and cecum at pH 8.5 for 20 min).

**Figure 7 viruses-13-01932-f007:**
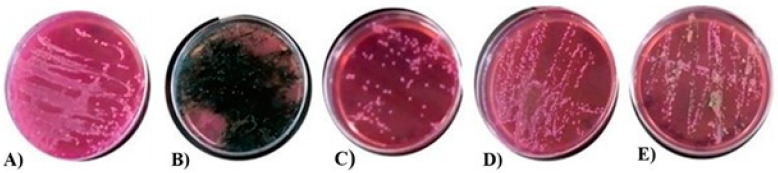
Reduction of growth SE PT13A in the cloacal samples obtained 48 h after the administration of the corresponding treatment. (**A**) Control group, uninfected and non-treated chickens; group (**B**) chickens infected with SE PT13A without treatment; group (**C**) chickens infected with SE PT13A and treatment with antibiotics; group (**D**) chickens infected with the bacteria and treatment with 1 g of beads containing phage; and group (**E**) chickens infected with SE PT13A and treatment with beads without phage.

**Table 1 viruses-13-01932-t001:** Infection registration and types of treatments in each experimental group.

Group	Infection			Treatments
	Day 1	Day 2	Day3	Day 8
A	No infection			ALG beads + S1 at 4.9 × 10^9^ PFU
B	S.E. at 4 × 10^6^ UFC	S.E. at 4 × 10^7^ UFC	S.E. at 2.5 × 10^9^ UFC	No treatment
C				Antibiotic treatment
D				ALG beads + S1 at 4.9 × 10^9^ PFU
E				ALG beads

S.E., Salmonella Enteritidis (SE PT13A). S1, phage S1. ALG, alginate. All ALG treatments weighted 1 g of mass. Antibiotic treatment indicates a mixture of 8 g of Trimethoprim/Sulfamethoxazole in 4 L of water. *n* = 30 chickens, 6 per group.

**Table 2 viruses-13-01932-t002:** Bacteriophage S1 ORFs with known and putative functions.

ORF	Product	Chromosomal Locus (nt)	Accession Number	Database	*e-*Value
1	Major capsid protein	1–978	YP009821716.1	NCBI	5.2 × 10^213^
6	Putative tail protein 1	2907–3527	QMV47842.1	NCBI	3 × 10^−134^
7	Putative tail protein 2	3527–6484	Q6QGE2	UniProtKB	0
8	Putative tail protein 3	6554–7195	Q6QGE2	UniProtKB	9.5 × 10^−10^
9	Putative tail protein 4	7208–9082	QMV47844.1	NCBI	6.8 × 10^−32^
10	Putative sheath protein	9170–10,309	Q24LI4	UniProtKB	5.6 × 10^−19^
11	Putative structural protein	10,320–10,751	D6RRG7	UniProtKB	1.8 × 10^−13^
12	Putative tail assembly chaperone protein	10,768–11,196	PF10876.10	Pfam	5.6 × 10^−15^
13	Putative tape measure protein	11,362–13,023	Q6KGH8	UniProtKB	7.2 × 10^−06^
16	Putative baseplate protein 1	14,250–15,200	10312	UniProtKB	1.8 × 10^−02^
17	Putative puncturing protein	15,190–15,834	PF18352.3	Pfam	5 × 10^−18^
18	Putative baseplate protein 2	15,843–16,214	P09425	UniProtKB	4.1 × 10^−7^
19	Putative baseplate protein 3	16,218–17,381	Q9T1V2	UniProtKB	3 × 10^−10^
21	Tail fiber protein S	18,020–19,369	Q9T1V0	UniProtKB	6.9 × 10^−18^
22	Tail fiber assembly protein U	19,369–19,911	Q71TD6	UniProtKB	2.9 × 10^−18^
23	Putative tail fiber assembly protein U’	19,914–20,423	Q71TD7	UniProtKB	6.9 × 10^−18^
24	Putative holin	20,525–20,788	PF16080.7	Pfam	6.7 × 10^−04^
25	SAR-endolysin	20,763–21,308	Q37875	UniProtKB	7.4 × 10^−13^
26	Putative spanin inner membrane subunit	21,287–21,616	Q9T1X1	UniProtKB	2 × 10^−9^
29	DNA polymerase	22,342–24,312c	P19822	UniProtKB	1.5 × 10^−49^
31	Thymidylate synthase	25,340–26,224c	P00471	UniProtKB	3.4 × 10^−40^
36	Putative ATP-dependent DNA helicase	28,586–30,259c	P20703	UniProtKB	2.3 × 10^−32^
38	Putative exonuclease	31,377–30,466c	P03697	UniProtKB	1.8 × 10^−04^
39	Putative deoxycytidylate deaminase	31,343–31,807c	P00814	UniProtKB	1.4 × 10^−13^
54	Anaerobic NTP reductase small subunit	38,921–41,455	YP009821771.1	NCBI	0
55	Putative resolvase	41,950–42,378	Q98VP9	UniProtKB	1.6 × 10^−04^
68	Putative dihydrofolate reductase	46,756–47,427	Q6QGJ4	UniProtKB	6.4 × 10^−06^
73	Putative terminase large subunit	49,484–50,932	P54308	UniProtKB	6.3 × 10^−16^
74	Portal protein	50,934–52,496	O64207	UniProtKB	6.5 × 10^−10^

Complementary strand, c.

**Table 3 viruses-13-01932-t003:** Efficiency of Plating (EOP) of phage S1.

Bacteria	Identification of Bacteria	EOP
*S.* Enteritidis (SE PT13A)	Host strain	1
*S.* Typhi	ICTH	0
*S.* Pullorum	ICTP	0.61
*S.* Gallinarum	ICTP	0.58
*S.* Cholerasuis	ICTH	0
*C. freundii*	ICTH	0
*E. cloacae*	ICTH	0
*E. coli*	ICTH	0

ICTP: Isolated from clinical trials in poultry, ICTH: Isolated from clinical trials in humans.

**Table 4 viruses-13-01932-t004:** Presence and absence of lytic zones of phage S1 in GIT of chickens after ALG beads administration.

Section of the GIT	1 H Post Infection		3 H Post Infection		5 H Post Infection		17 H Post Infection	
	C #1	C #2	C #1	C #2	C #1	C #2	C #1	C #2
Crop	** *+* **	** *+* **	**−**	**+**	**−**	**+**	**−**	**−**
Gizzard	**−**	**−**	**−**	**−**	**−**	**−**	**−**	**−**
Duodenum	**−**	**−**	**−**	**+**	**−**	**+**	**−**	**−**
Cecum	**−**	**−**	**+**	**+**	**−**	**+**	**−**	**−**

GIT, gastrointestinal tract. Chicken, C. +, Presence of lytic plaques. −, Absence of lytic plaques. *n* = from 38 chickens, 8 were randomized selected.

## Data Availability

Taken in part from the experimental work of the Ph. D. student Janeth Gomez-Garcia. The genome sequence of the bacteriophage S1 is available at the NCBI repository under the accession number MZ127825.

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
