# Peer review of "Efficacy of Salmonella Bacteriophage S1 Delivered and Released by Alginate Beads in a Chicken Model of Infection"

_viruses, 2021, doi:10.3390/v13101932_

Round 1

Reviewer 1 Report

In the introduction could be mentioned that bacteriophages in poultry could be used also to reduce salmonella in bird droppings which can reduce feedback contamination related to fertilization of the fields and improvement of food safety [Grygorcewicz, B., Grudziński, M., Wasak, A., Augustyniak, A., Pietruszka, A. and Nawrotek, P., 2017. Bacteriophage-mediated reduction of Salmonella Enteritidis in swine slurry. Applied Soil Ecology119, pp.179-182.; Grygorcewicz, B., Chajęcka‐Wierzchowska, W., Augustyniak, A., Wasak, A., Stachurska, X., Nawrotek, P. and Dołęgowska, B., 2020. In‐milk inactivation of Escherichia coli O157: H7 by the environmental lytic bacteriophage ECPS‐6. Journal of Food Safety40(2), p.e12747.]. 

with respect to methodological section 2.10.3. Elimination of SE PT13A in chickens - why authors do not serial dilution and plating? this will add more valuable data?

Reviewer 2 Report

The approach of this manuscript is interesting for the application of Phage therapy in animals, which is often not so restricted than for humans. Therefore, it is important to explore and demonstrate the action of phages in clinical applications, even if it is in animals.

I consider this manuscript suitable for publication, after the authors address the minor issues outlined below:

  • Line 3: remove the final dot at the end of the title.
  • Line 28: “…formulation is an effective system…”
  • Line 29: “delivery of phage…”
  • Lines 48-49: do you mean “demonstrated” instead of demonstrating?
  • Line 51: “…bacteriophages to reduce…”
  • Line 91: “…México City, Mex)”, in line 90 you wrote “(BD Bioxon, Mexico City, México)”, please be uniform. Also, you mention “México” and ”Mexico”, please be uniform.
  • Line 113: “…was poured…”
  • Line 149: “…the EOP in was…”, remove the “in”
  • Line 158: in the formula you wrote “pjaques” instead of plaques.
  • Line 180: “…S1 that is not…”, remove “is” or wrote “was not”
  • Line 184: “made. using” remove the “.”
  • Lines 203-204: this subtitle is too long, please shorten it. Suggestion: “In vitro protection and release of free phages and encapsulated phages in ALG beads”. Or just “In vitro assays”.
  • Line 222: the end of this sentence is confused. Remove “or their disintegration due to” and write “as the ALG beads disintegrate…”
  • Line 283: in Table 1 remove the “.” after “Group”.
  • Line 284: to be consistent put “S1, Bacteriophage S1”.
  • Line 287: remove the “.” after “Results and Discussion”.
  • Line 332: “sharing “instead of “shared”.
  • Line 346: “noticed” instead of “notice”.
  • Line 358: “…only 57 were found…”
  • Line 374: in line 330 you wrote “Loughboroughvirus” in italic, here it is not in italic.
  • Line 377: “sharing “instead of “shared”.
  • Line 400: “...showed that an average of..” Remove “that”.
  • Line 407: why are chitosan and polyethylenimine in capital letters?
  • Lines 421-422: remove “the bacterial growth”, it is repeated.
  • Line 425: in the legend of the bars of figure 5 remove “with” just write “ALG” and “ALG + phage”.
  • Lines 428-429: the same I wrote for lines 203-204. It is a very long subtitle.
  • Line 449: “release” instead of “releasing”.
  • Line 450: remove “which”.
  • Line 499: “…in all groups, the presence of lactose-positive…”

Along the text you mention “phages” and “bacteriophages”, once you explain that phages is the short term for bacteriophages you can use only phages. Otherwise, just choose one of the terms and do not alternate between both. Moreover, you wrote titer and titre, choose just one.

In the Host range by EOP (subtitle 3.2), I miss some discussion about those values presented in table 3, why do you think that happened?

All figures lack good resolution.
